# Transformers Have the Potential to Achieve AGI

## Abstract

As large language models (LLMs) based on the Transformer architecture continue to achieve impressive performance across diverse tasks, this paper explores whether Transformers can ultimately achieve artificial general intelligence (AGI). We argue that Transformers have significant potential to achieve AGI, supported by the following insights and arguments. (1) A Transformer is expressive enough to simulate a programmable computer equipped with random number generators and, in particular, to execute programs for meta-tasks such as algorithm design. (2) By the extended Church-Turing thesis, if some realistic intelligence system (say, a human with pencil and paper) achieves AGI, then in principle a single Transformer can replicate this capability; Besides, we suggest that Transformers are well-suited to approximate human intelligence, because they effectively integrate knowledge and functions represented in network form (e.g. pattern recognition) with logic reasoning abilities. (3) We argue that Transformers offer a promising practical approximation of Hutter's AIXI agent, which is an ideal construction to achieve AGI but is uncomputable.

## 1 Introduction

Large language models (LLMs) [1–4] have demonstrated remarkable capabilities across a broad range of challenging tasks. For example, OpenAI's o-series [5] model achieves 71.7% accuracy on the software engineering benchmark SWE-bench [6], 87.7% on the graduate-level question answering task GPQA [7], and 96.7% on a competition-level mathematics reasoning task [8]. Notably, these results surpass human-expert performance. As LLMs evolve, their capabilities are expected to advance further.

These successes are grounded in the Transformer architecture [9], which has proven to be highly effective across a wide range of domains, extending beyond natural language processing to areas such as computer version [10] and decision-making [11]. Given the impressive achievements of Transformers in tackling challenging tasks across various domains, a fundamental question arises:

***Question 1****: Can Transformers ultimately achieve artificial general intelligence (AGI)?*

To answer this question, we must first establish a rigorous definition of intelligence. Intelligence is multifaceted, encompassing abilities such as creativity, problem-solving, pattern recognition, and reasoning. However, formulating a single, comprehensive definition that captures all these aspects is challenging. As pointed out in [12], most, if not all, aspects of intelligence can be framed in terms of goal-driven behavior, or more precisely, as the maximization of some (often unknown) utility (reward) function. This aligns with the "reward is enough" hypothesis [13], which suggests that the pursuit of maximizing reward alone is sufficient to drive behaviors that exhibit a wide range of capabilities, many of which are traditionally studied in both natural and artificial intelligence.

In this paper, we follow the definition that intelligence can be broadly categorized into two types of reasoning abilities:

Submitted to 39th Conference on Neural Information Processing Systems (NeurIPS 2025). Do not distribute.

- Learning an unknown utility function (inductive reasoning): This involves drawing generalizations from specific observations, where the conclusions are probable but not certain. This type of reasoning is extensively explored in the context of inverse reinforcement learning [14, 15]. Examples of inductive reasoning include pattern recognition, natural language processing, prediction, and scientific research, where repeated observations lead to hypotheses or theories.

- Maximizing a known utility function (deductive reasoning): In this case, the solution depends entirely on the explicit, provided information. Successful applications includes AlphaGo [16], Muzero [17], AlphaProof [18], OpenAI-o1 [5], and DeepSeek-R1 [19].

In this paper, we argue in favor of Question 1, **supporting the potential of Transformers to achieve AGI** with the following insights and arguments.

**1. A single Transformer can simulate a probabilistic programmable computer.** Prior works (e.g. [20]) have shown that Transformers (with chain-of-thoughts) can efficiently simulate deterministic Turing machines (DTMs). We extend this result to the potentially more powerful probabilistic Turing machines (PTMs), proving that Transformers can efficiently simulate PTMs as well (Theorem 2).

At first glance, Theorem 2 may suggest adherence to a one-model-one-task paradigm, where different tasks require different transformers. This misaligns with the current practice of training a single general-purpose transformer to perform various tasks. In fact, Theorem 2 provides deeper insights: as also observed in related work (e.g. [21]), it implies that a *single* Transformer can simulate a probabilistic universal Turing machine (UTM), a formalization of a general-purpose programmable computer equipped with random number generators.

Furthermore, while Transformers do not follow the one-model-one-task paradigm, they appear to adhere to a one-prompt-one-task paradigm, where different tasks require different PTMs (or equivalently, programs) to be specified in the prompt or pre-injected during training. We argue that this is *not* the case. Specifically, beyond algorithms for specific tasks, a PTM $T$ can also serve as a program for meta-tasks, such as designing other algorithms (meta-algorithms), or even higher-order tasks, such as meta-meta-algorithms.

**2. Implication of the extended Church-Turing thesis.** The *extended Church-Turing thesis* (ECT) [22, 23], an extension of the Church-Turing thesis in the modern computer science literature from a complexity-theoretic perspective, asserts that the PTM model is not only as expressive as but also as efficient as any realistic physical device (say, a human brain, a society, or a future neural network). Specifically, any function that can be computed by a realistic finite physical system can also be computed by a PTM with at most a polynomial slowdown. Consequently, if some realistic intelligence system (say, a human brain with pencil and paper) achieves AGI, then in principle, a single Transformer can achieve AGI as well (Thesis 1).

In particular, Thesis 1 suggests that a single Transformer has the potential to achieve human-level intelligence. Moreover, we suggest that Transformers are particularly well-suited as approximations of human intelligence, because they effectively integrate knowledge and functions represented in network form with logical reasoning abilities, and thus can leverage benefits from both connectionism AI and symbolicism AI.

**3. Algorithmic approximations of general intelligence.** Besides mimicking the human reasoning process, another line of research, inspired by algorithmic information theory, seeks to reach or even outperform human intelligence by establishing a formal theory of general intelligence. Several constructions have been proposed to address meta-tasks, including:

*Levin's universal search algorithm*: Many deductive reasoning tasks, such as theorem proving, planning, and general NP-complete problems, can be effectively modeled as search problems. Levin's universal search is an algorithm that can solve all search problems as quickly as the fastest algorithm for each, up to a large constant factor [24, 25]. The basic idea is to run all programs $p$ in parallel with relative computation time $2^{-\ell(p)}$; i.e. a time fraction $2^{-\ell(p)}$ is dedicated to executing $p$. Here, we describe programs as Boolean strings using a prefix-free encoding, where $\ell(p)$ denotes the length of the description of $p$. Note that the sum of all these time fractions satisfies $\sum_p 2^{-\ell(p)} \leq 1$.

*Solomonoff's universal induction*: Every inductive reasoning task, such as continuing a number of series in an IQ test, classification in machine learning, stock-market forecasting, or scientific research, can be described as a sequence prediction problem, more precisely, predicting future data from past observations [12]. Solomonoff's universal induction [26, 27] is an optimal approach for all

sequence prediction problems, where the data is sampled from a computable probability distribution, or equivalently generated by a realistic physical system according to the physical version of Church-Turing thesis. The basic idea is to do Bayesian prediction, using Solomonoff prior as the prior belief, which assigns higher probabilities to simpler hypotheses with shorter descriptions, aligning with Occam's razor.

*Hutter's AIXI agent*: AIXI is a theoretical agent that achieves AGI [12]. AIXI is somehow a combination of Solomonoff's universal induction and Levin's universal search. Specifically, AIXI replaces the unknown environment in the Bellman equation with a generalized Solomonoff prior and then invokes $M_{p^*}^\epsilon$ [12], an enhancement of Levin's universal search, to solve the Bellman equation. Like Solomonoff induction, AIXI tends to hypothesize the environment as shortest possible programs, in line with Occam's razor.

While these constructions are theoretically optimal, they are often intractable in practice or even uncomputable. However, we argue that Transformers provide a promising tractable approximation of these universal constructions. Specifically,

*Universal search*: By Theorem 2, a single Transformer, by simulating Levin search, can theoretically solve all search problems as efficiently as the fastest algorithm for each problem. To enhance tractability, Transformers can leverage prior knowledge embedded during training to assign the relative computation time proportion in a more adaptive and efficient way. In addition, Transformers can continually refine search strategies by learning from past experiences.

*Universal induction*: Recent works [28–32] have demonstrated that Transformers align with Occam's razor, the core principle of Solomonoff induction. Specifically, Transformers tend to output sequences generated by shorter programs (a.k.a. with lower Kolmogorov complexity). The alignment with Occam's razor enables Transformers to generalize effectively across diverse tasks and data modalities, making them good approximations of general-purpose predictors. Furthermore, [32] put forth and explore a hypothesis that Transformers approximate Solomonoff induction better than any other extant sequence prediction method, highlighting their potential as practical implementations of universal induction.

*AIXI agent*: we suggest that Transformers have the potential to offer a practical approximation of AIXI for the following reasons: as we just discussed, (i) Transformers have the potential to approximately implement Solomonoff universal induction; (ii) Transformers has potential to implement universal search in practice, enabling efficient solutions for a wide range of deductive reasoning tasks; and (iii) Transformers integrate prior knowledge effectively, leveraging human experience to enhance their practical applicability. Moreover, for practicality, we propose a framework that approximates the AIXI agent using two complementary Transformers in Section 4.3.

## 2 Transformers can efficiently simulate probabilistic TMs

We assume that the reader is familiar with the definitions of the Transformer architecture, Turing machine (TM), probabilistic Turing machine (PTM), and universal Turing machine (UTM). For the convenience of readers, we present a background of TM in Appendix A. There is a line of theoretical works [33–37, 20, 38, 39, 21] studying the expressive power of Transformer with chain of thought (CoT) by connecting them with Turing machines. It turns out that decoder-only Transformers with $t$ CoT steps can simulate $t$ DTM steps.

**Theorem 1** ([20]). *Let $T$ be a deterministic Turing machine that, on input $x$ of length $n$, runs for at most $t(n)$ steps. There is a constant-depth decoder-only Transformer that, on input $x$, takes $t(n)$ CoT steps and then outputs $T(x)$.*

In this paper, we extend Theorem 1 to PTMs. In particular, it implies that Transformers with polynomial CoT steps can solve all problems in BPP, the class of decision problems solvable by a PTM in polynomial time, which is *strictly* larger than the P class, which consists of all decision problems solvable by a DTM in polynomial time, unless BPP = P. The proof of Theorem 2 can be found in Appendix B.

**Theorem 2.** *Let $T$ be a probabilistic Turing machine that, on input $x$ of length $n$, runs for at most $t(n)$ steps. There is a constant-depth decoder-only transformer that, on input $x$, takes at most $2t(n)$ CoT steps and returns the same (randomized) output as $T$.*

At first glance, Theorem 2 appears to follow the one-model-one-task paradigm: different tasks require different Transformers. This misaligns with the current practice of training a single general-purpose transformer to perform various tasks. In fact, Theorem 2 provides deeper insights: as also observed in related works (e.g. [21]), Theorem 2 implies that a *single* Transformer can simulate a UTM, or equivalently a programmable computer equipped with random number generators.

Though Transformers do not have to follow the one-model-one-task paradigm, they appear to follow the one-prompt-one-task paradigm: for different tasks, different PTMs should be loaded into the prompt or pre-injected during training. We argue that this is not the case. Specifically, beyond algorithms for specific tasks, the PTM $T$ can also be taken as a program that performs meta-tasks, such as a meta-algorithm that designs algorithms, or even meta-meta-algorithms. For example, the Transformer can implement the following meta procedure: it takes a problem description and an input $x$ as input, and then

1. run some prescribed meta-algorithm to initialize or update a program $p$;
2. run $p$ on input $x$, and obtain $p(x)$;
3. evaluate $p(x)$. If not good enough, then go to Step 1.

# 3 The extended Church-Turing thesis and its implication

The physical version of *Church-Turing thesis* (CT), as known as Deutsch-Wolfram thesis, asserts [40–42] that every finite physical system (say, a modern personal computer, a human brain, a society, or a future neural network) can be simulated to any specified degree of accuracy by a PTM. Furthermore, there is also a strengthening, referred to as the *Extended Church-Turing thesis* (ECT), of the physical Church-Turing thesis in the modern computer science literature [22, 23] from a complexity-theoretic perspective, asserting that the probabilistic Turing machine model is also as efficient as any computing device can be. That is, if a function is computable by some hardware device in time $T(n)$ for the input of size $n$, then it is computable by a PTM in time $O(T(n)^k)$ for some constant $k$.

**Remark 1.** *The physical version of CT and ECT are very different from the original version proposed by Church and Turing [43, 44], which asserts that every algorithmic process can be carried out by a PTM. Specifically, if a task can solved by a human being with paper and pencil by following a finite number of exact instructions, then the original CT asserts that it can also be solved by a PTM. Notably, no insight, intuition, or ingenuity is demanded on the part of the human being carrying out the method, which is very different from the physical version. The original CT is something between a theorem and a definition. And the physical version and ECT are neither mathematical theorems nor definitions. If they are true, then the truth is a consequence of the laws of physics [45].*

By combining Theorem 2 and ECT, we obtain the following thesis:

**Thesis 1.** *If some realistic intelligence system (say, a human brain with pencil and paper) achieves AGI, then a single Transformer can also achieve AGI with at most a polynomial slowdown.*

## 3.1 Algorithmically description of human reasoning

If we accept that (a) the extended Church-Turing thesis applies to the human's reasoning process, meaning that the reasoning process of humans can be efficiently simulated by a PTM, and (b) a human brain, or a group of human brains (say, a research community) with paper and pencil can achieve AGI, then by Theorem 2, we should also accept that in principle a single Transformer or a group of Transformers can also achieve AGI as well. The related challenge lies in algorithmically describing the human reasoning process, including cognitive functions like intuition or creativity.

*Question 2: How to algorithmically describe human reasoning process?*

There are two kinds of general approaches to this challenge: connectionism and symbolicism.

- **Connectionism: Simulating the Brain at the Physical Level.** Connectionism posits that human reasoning arises from the emergent properties of biologically inspired neural networks. By modeling the brain's physical and biological substrates—specifically, the interactions of neurons through synaptic connections—this approach seeks to replicate cognitive processes via distributed, parallel computation. Modern artificial neural networks, such as deep learning architectures,

exemplify this paradigm. These systems learn hierarchical representations from data, mirroring how the brain processes sensory input and abstracts patterns [46, 47].

For instance, Transformer architectures [9] model sequential reasoning by leveraging temporal dependencies and attention mechanisms, achieving expert-level performance in complex mathematical reasoning and code generation tasks [5, 48]. Connectionist models excel at pattern recognition and probabilistic reasoning but often lack explicit symbolic representations, leading to critiques about their interpretability and inability to handle structured, rule-based logic [49]. Recent advances in neuro-symbolic integration, however, aim to bridge this gap by combining neural networks with symbolic reasoning modules [50, 51].

- **Symbolism: Abstracting General Principles of Human Thought.** Symbolism adopts a top-down perspective, seeking to formalize the universal principles and logical structures that underpin human reasoning. Rooted in classical AI and influenced by philosophy, linguistics, and formal logic, this approach abstracts cognition into discrete symbols and rules, independent of biological implementation. Unlike connectionism, which emulates neural substrates, symbolism prioritizes computational-level explanations of thought—asking what problems cognition solves and why, rather than how the brain physically solves them [52, 53].

  At its core, symbolism assumes that reasoning can be modeled as manipulation of explicit representations through deterministic or probabilistic rules. For example: Occam's razor, a heuristic for inductive reasoning, is formalized in algorithmic frameworks like Bayesian model selection [54], where simpler hypotheses are assigned higher prior probabilities. Deductive reasoning is captured by logic-based systems (e.g., Prolog, theorem provers) that apply syllogistic rules (e.g., modus ponens) to derive conclusions from premises [55].

Here, we argue that an effective solution requires a combination of these two approaches, since (i) abstracting general principles offers a more tractable and generalizable framework for intelligence and (ii) part of knowledge and functions, such as pattern recognition and cognitive functions, may have no representation more concise than a huge, analog neural network [56], thus are not suitable to be represented as logic or symbolic.

In particular, since Transformers can effectively integrate knowledge and functions represented in network form (since they are neural networks) with logical reasoning abilities (Theorem 2), and thus can leverage benefits from both connectionism and symbolism, we suggest that Transformers are particularly well-suited as approximations of human intelligence.

# 4 Algorithmic approximations of general intelligence

Besides mimicking human reasoning process, another line of research, inspired by algorithmic information theory [57], aims to achieve or even surpass human-level intelligence by establishing a formal theory of general intelligence, such as Levin's universal search algorithm [24, 25], Solomonoff's universal induction [26, 27], and Hutter's AIXI agent [12]. While these constructions are theoretically optimal, they are often intractable in practice and even uncomputable. However, we argue that Transformers provide a promising and tractable approximation of these universal constructions.

## 4.1 Levin's universal search

Many deductive reasoning tasks, such as theorem proving, planning, and general NP-complete problems, can be effectively modeled as search problems.

**Search problems.** Let $\phi : \{0, 1\}^* \rightarrow \{0, 1\}^*$ be a function where $\phi(\cdot)$ can be computed quickly (say, in polynomial time). The search problem is defined as: given $y$, find an $x$ such that $\phi(x) = y$.

For example, in the Boolean satisfiability problem (SAT), the function $\phi : \{0, 1\}^* \rightarrow \{0, 1\}$ can be defined as a verifier that checks whether a given assignment satisfies the Boolean formula in conjunctive normal form.

**Levin search.** The algorithm is simple to describe: just run and verify the output of all algorithms $p$ in parallel with relative computation time $2^{-\ell(p)}$; i.e. a time fraction $2^{-\ell(p)}$ is devoted to executing $p$ [24, 25]. Here, programs are described as Boolean strings in a prefix-free encoding, where $\ell(p)$ denotes the length of the description of $p$. Note that $\sum_p 2^{-\ell(p)} \leq 1$.

**Theorem 3** ([24, 25, 12]). *The computation time of Levin search is upper bounded by* $\min_p\{2^{\ell(p)} \cdot \text{time}_p^+(y)\}$*, where* $\text{time}_p^+(y)$ *is the runtime of* $p(y)$ *plus the time to verify the correctness of the result* ($\phi(x) = y$) *by a known implementation for* $\phi$.

By Theorem 2, we conclude that in principle, a single Transformer can solve all search problems as quickly as the fastest algorithm for each, up to a constant factor.

We note that Levin's universal search—which optimally allocates computational effort across candidate solvers according to their algorithmic probability (Theorem 3)—may provide a theoretical foundation for the emerging paradigm of inference-time scaling in LLMs [58, 59, 5, 48]. This framework structures LLM reasoning into two synergistic phases: generating diverse candidate solutions (or algorithms) and efficiently prioritizing their execution and evaluation, mirroring Levin's time-optimal balance between exploration and exploitation. By prescribing a focus on programs with minimal description length (i.e., favoring simpler, valid solutions), Levin's principles offer guidance for designing compute-efficient strategies.

Though Levin search is theoretically optimal for all search problems, the large constant overhead $2^{\ell(p)}$ renders it impractical. A line of research [60–64] has explored adaptations of Levin's search that leverage past experience to improve its efficiency. We note that the key lies in generating highly successful algorithms $p$ with the shortest description length, ensuring they are prioritized during the search process. Such knowledge can be acquired from experience. For instance, a Transformer could maintain a parameterized model (e.g., a neural network or program) within its context and employ bootstrap methods—such as search-and-learn processes [65]—to iteratively refine its performance. By repeatedly solving increasingly challenging instances and updating the model based on successfully solved examples, the system could incrementally improve its problem-solving efficiency.

## 4.2 Solomonoff's universal induction

Every inductive reasoning task, such as continuing a number of series in an IQ test, classification in machine learning, stock-market forecasting, or scientific research, can be described as a sequence prediction problem, more precisely, predicting future data from past observations [12]. Without loss of generality and for simplicity, we assume the data $x_i \in \{0, 1\}$ is binary.

We first introduce some notations and definitions. Given a subset $S$ of $\{0, 1\}^\star$, let $\lfloor S \rfloor$ denote the set obtained from $S$ by deleting all elements that have a prefix in $S$. A *monotone Turing machine* is a Turing machine with one unidirectional ready-only input tape, one unidirectional write-only output tape, and some bidirectional work tapes. We say a tape is unidirectional if its head can only move from left to right, and bidirectional if its head can move in both directions.

**Definition 1** (Measure). *We say a function* $\mu : \{0, 1\}^* \to [0, 1]$ *is a measure if* $\mu(\emptyset) = 1$ *and* $\mu(x) = \mu(x1) + \mu(x0)$. *Here,* $\emptyset$ *denotes the empty string.*

*A measure* $\mu$ *defines a random process generating an infinitely long binary sequence: start with an empty string and repeatedly select the next bit* $x_n \in \{0, 1\}$ *according to the probability* $\mu(x_n \mid x_{<n}) := \mu(x_{<n}x_n)/\mu(x_{<n})$ *conditioned on the past data* $x_{<n} := x_1 x_2 \cdots x_n$.

We say $\mu$ is estimable if there exists a TM that, given $x \in \{0, 1\}^*$ and a precision $\epsilon$, computes an $\epsilon$-approximation of $\mu(x)$. By the physical version of the Church-Turing thesis, any $\mu$ implemented on a finite, realistic physical device is estimable. Moreover, by Theorem 4.5.2 in [57], for any estimable $\mu$, there is a monotone TM $T$ that takes an infinitely long uniformly random binary string as input and generates an infinitely long binary sequence according to $\mu$. Let $K(\mu)$ denote the shortest description of such a $T$.

**Sequence prediction problem.** Having observed the past data $x_{<n} := x_1 x_2 \cdots x_{n-1}$, the task is to predict the next bit $x_n$. More precisely, let $\mu$ denote the unknown underlying mechanism generating the sequence $x_1 x_2 \cdots$. The task is to estimate the conditional probability $\mu(x_n \mid x_{<n}) := \mu(x_{<n}x_n)/\mu(x_{<n})$.

**Solomonoff's universal induction.** Bayesian prediction provides a framework for sequence prediction problems, which repeatedly employs Bayes' rule to update its beliefs about each hypothesis based on newly observed data. The primary challenge is how to select the prior beliefs. Solomonoff [26, 27] addressed this challenge by introducing a universal prior, rooted in the simplicity of hypotheses. His approach leverages the fact that simpler hypotheses, represented by shorter programs, are more likely

to generalize well—a concept aligned with Occam's Razor. Solomonoff showed that the Bayesian prediction with the Solomonoff prior as the prior belief is an optimal way for the sequence prediction problem, provided that the underlying $\mu$ is estimable.

**Definition 2** (Solomonoff prior [26, 27]). *Let $U$ be a monotone UTM. The Solomonoff prior is defined as*

$$M_U(x) := \sum_{\lfloor p \in \{0,1\}^* : U(p)=x\star \rfloor} 2^{-\ell(p)}.$$

*Here, $U(p) = x\star$ means $x$ is a prefix of $U(p)$. Intuitively, $M_U(x)$ is the probability that the output starts with $x$ when the input is an infinite-long uniformly random binary string.*

Solomonoff's universal induction (SI) is simple to describe: use $M_U(x_n \mid x_{<n}) = M_U(x_n)/M_U(x_{<n})$ as an estimate of the true conditional probability $\mu(x_n \mid x_{<n})$.

**Theorem 4** (Solomonoff central theorem [26, 27]). *For any estimable $\mu$, we have*

$$\sum_{n=1}^{+\infty} \sum_{x_n \in \{0,1\}} \mu(x_{<n}) \left(M(x_t \mid x_{<t}) - \mu(x_n \mid x_{<n})\right)^2 \leq \ln 2 \cdot K(\mu) + O(1).$$

For any estimable $\mu$, the upper bound $\ln 2 \cdot K(\mu)$ is finite, so the difference $M(x_t \mid x_{<t}) - \mu(x_n \mid x_{<n})$ tends to zero as $n \to \infty$ with $\mu$-probability 1. Consequently, $M(x_t \mid x_{<t})$ converges rapidly to the true underlying generating process.

Unfortunately, Solomonoff prior $M_U(x)$ is inestimable: there is no TM that, given $x \in \{0,1\}^*$ and a precision $\epsilon$, can compute an $\epsilon$-approximation of $M_U(x)$ in finite time. To address this uncomputability issue, several approximations have been proposed [66–68, 30].

In particular, observing that Transformers are naturally suited for sequence prediction tasks, a line of work [28–32] has explored whether the Transformer model can approximate SI. Specifically, [29, 28] showed that transformers can do Bayesian inference. [30] proved that Transformers can approximate SI by training solely on UTM data, and demonstrated that increasing model size leads to improved performance. [32] proposed and investigated the hypothesis that Transformers approximate Solomonoff induction better than any other extant sequence prediction method. This hypothesis was further supported by [31, 69]. Specifically, they showed that like Solomonoff induction, transformers also align with Occam's Razor: transformers prefer generating data with low Kolmogorov complexity. Occam's razor provides transformers with good generalization on many different problems and modalities of data, and makes them powerful general-purpose predictors.

### 4.3 Hutter's AIXI agent

We first briefly introduce Hutter's AIXI agent, which is claimed to be universal in that it is independent of the true environment (model-free) and is able to solve any solvable problem and learn any learnable task. The main idea of AIXI is simple to describe: just replace the unknown environmental distribution in the Bellman equations with a suitably generalized Solomonoff prior [12].

**Setting.** The agent and the environment interact chronologically as follows: in each cycle $k$, the agent performs an action $y_k \in \mathcal{Y}$ (output), and then receives a perception $x_k \in \mathcal{X}$ from the environment. The perception $x_k$ consists of a regular part $o_k$ and a reward $r_k$. Given the history $y_1 x_1 \cdots x_{k-1} y_k$, the probability that the environment produces perception $x_k$ is denoted $\mu(x_k \mid y_1 x_1 \cdots x_{k-1} y_k)$. Here, we make no assumptions about $\mu$ other than it is estimable. In particular, $\mu$ is allowed to depend on the complete history $y_1 x_1 \cdots x_{k-1} y_k$.

We use $p$ to denote the agent's policy, which can be described as a monotone Turing machine that takes $x_1 x_2 \cdots$ as input and outputs $y_1 y_2 \cdots$. As the optimal policy can always be chosen to be deterministic, we assume $p$ is a deterministic monotone TM. In addition, we say $y_{1:k} = p(x_{<k})$ if $y_i = p(y_1 x_1 y_2 x_2 \cdots x_{i-1})$ for $i \leq k$. We also use $\mu(x_{k:m} \mid y_{1:m} x_{<k})$ as an abbreviation for $\Pi_{i=k}^m \mu(x_i \mid x_{<i}, y_{\leq i})$. We define the value of policy $p$ in environment $\mu$ as

$$V_\mu^p := \sum_{x_{1:m}} (r_1 + \cdots + r_m) \mu(x_{1:m} \mid y_{1:m})_{|y_{1:m}=p(x_{<m})}$$

where $m$ is the lifespan of the agent.

The goal of the agent is to maximize the total reward $\sum_{i=1}^m r_i$. Formally, the agent aims to find a policy $p^\mu$ that maximizes $V_\mu^p$.

**The AIXI agent**. If the environment $\mu$ is known, then the optimal policy is

$$y_k := \arg\max_{y_k} \sum_{x_k} \cdots \max_{y_m} \sum_{x_m} \left( \sum_{i=k}^{m} r_i \right) \mu(x_{k:m} \mid y_{1:m}, x_{<k})$$

with total reward

$$\max_{y_1} \sum_{x_1} \cdots \max_{y_m} \sum_{x_m} (r_1 + \cdots + r_m) \mu(x_{1:m} \mid y_{1:m}) := V_\mu^*$$

The AIXI agent replaces the true but unknown $\mu$ with a generalized Solomonoff prior. Specifically, the AIXI policy is

$$y_k := \arg\max_{y_k} \sum_{x_k} \cdots \max_{y_m} \sum_{x_m} \left( \sum_{i=k}^{m} r_i \right) \xi(x_{k:m} \mid y_{1:m}, x_{<k})$$

where

$$\xi(x_{1:k} \mid y_{1:k}) := \sum_{\text{monotone TM } q:q(y_{1:k})=x_{1:k}} 2^{-\ell(q)}.$$

Intuitively, the agent continually updates its belief about hypotheses of the unknown environment $\mu$ by Bayes' rule. Similar to Solomonoff universal induction, environments with lower Kolmogorov complexity are preferred, in line with Occam's razor. [12] shows that AIXI's environment model converges rapidly to the true environment, and its policy is Pareto-optimal and self-optimizing. Here, we say a policy Pareto-optimal if there is no other agent that performs at least as well as AIXI in all environments while performing strictly better in at least one environment, and self-optimizing if $\frac{1}{m} V_\mu^{\text{AIXI}} \to \frac{1}{m} V_\mu^*$ for horizon $m \to +\infty$ for all estimable $\mu$.

Unfortunately, like Solomonoff's universal induction, AIXI is uncomputable. To address this issue, several computable approximations have been proposed [12, 70–72, 67, 73–77]. One such approximation is AIXI$t\ell$, which performs at least as well as any other agent bounded time $t$ and length $\ell$. Some approximations focus on restricted environment classes and have been successfully implemented [72]. [77] studied how to inject knowledge into the AIXI agent.

We suggest that Transformers have the potential to approximate AIXI for the following reasons: (i) as discussed above, Transformers might serve as a good approximation of SI, and provide a good estimation of $\xi$; (ii) with an estimation of $\xi$, Transformers can solve the Bellman equation using an enhanced Levin search or an enhanced $M_{p^*}^\epsilon$ algorithm [12], which solves all well-defined problems as quickly as the fastest algorithm for each; and (iii) Transformers effectively integrate prior knowledge, leveraging human experience to further enhance their practical applicability.

For practicality, we propose a framework that approximates AIXI using two complementary Transformers: a Environment Modeler (induction) and a Action Planner (deduction).

- **Transformer I (induction component): Environment Modeler.** This component aims to approximate SI, providing a good estimation of $\xi$. We list some potential methods to improve its practical inductive reasoning ability.

  - *Mixing synthetic UTM data with real-world data:* Training exclusively on synthetic UTM data is already sufficient to enable Transformers to converge to SI, as such data spans a universal distribution over computable sequences [30]. Nevertheless, by incorporating real-world data, we can leverage prior knowledge to make this component much more practical. Specifically, adding real-world data biases the model toward assigning higher probability to programs that are relevant to real-world environments. This leads to faster learning and fewer errors when the model is deployed in such environments, while still preserving the universality [30].

  - *Data and model scaling are essential:* As shown in [30], increasing model size leads to better approximation of SI. This supports the need for massive data and extensive pretraining.

  - *Leveraging environment feedback:* After deployment, the model can continually refine its prior through periodic fine-tuning on new feedbacks. While ICL is theoretically sufficient, practical Transformers are limited by finite context lengths, so parameter updates through finetuning are crucial to consolidate the information from the long-term feedback.

- **Transformer II (deduction component): Action planner.** This component solves the explicit decision-making or planning task based on the prediction from Transformer I. This is not a

machine learning problem by itself; so unlike the inductive component, massive training data are not required in principle.

Importantly, our deduction framework for the AIXI agent offers a promising direction for improving LLMs' test-time scaling behavior. While current methods employ direct reward maximization (e.g., RL) to drive reasoning—focusing narrowly on final outcomes—our approach integrates Levin search to systematically evaluate reasoning steps based on their algorithmic complexity. This principled methodology achieves an optimal balance between: exploration (discovering diverse reasoning pathways) and exploitation (optimizing high probability steps toward solutions). By formalizing this trade-off, our framework may provide a foundation for developing next-generation reasoning models capable of more sophisticated and scalable problem-solving.

Although our deduction component does not require training in principle, its practical performance can be significantly improved using the learning-to-optimize techniques [78]. For example, by leveraging human experience or prior problem-solving data, one can train a model that takes a problem description as input and outputs a learned prior over candidate programs. This learned prior is expected to assign higher weights to more promising algorithms, effectively replacing the default weighting in Levin's universal search.

Due to the space constraint, the discussion about the data requirement is deferred to Appendix D

## 5 Alternative Views

This section discusses alternative views arguing that Transformers are not a sufficient path to AGI.

*Alternative view 1: Transformers miss essential capabilities for intelligent beings, such as understanding and reasoning about the physical world. Specifically, Transformers cannot anchor their understanding in reality: They cannot perform actions in the real world or learn through embodied experiences, and they lack the capability for hierarchical planning, a crucial element for understanding and interacting with the world at multiple levels of abstraction (e.g. [79]).*

We acknowledge that current Transformers lack the capabilities to interact with the physical world directly and employ embodied learning. However, we do not think this represents an inherent limitation of Transformers. With minor enhancements, Transformers could be embedded within agent models. Specifically, such agents could utilize Transformers as an approximation of universal induction (Section 4.2) to learn about the unknown environment, and subsequently apply them as an approximation of universal search (Section 4.1) to perform deductive reasoning.

Besides, we argue that current Transformers really do understand. In our definition of intelligence (Section 1), understanding can be equated to inductive reasoning, i.e., the ability to uncover the underlying general mechanism from specific observations. As we argued in Section 4.2, Transformers provide a promising practical approximation of Solomonoff's universal induction, which is a universal and optimal way to do inductive reasoning.

In addition, as we argued in Section 2, Transformers can execute any meta-process, such as algorithm design, when an algorithmic description is provided. In particular, as argued in Section 4.1, Transformers provide a promising practical approximation of Levin's universal search algorithm, enabling them to efficiently perform various deductive reasoning tasks, including planning and theorem proof.

*Alternative view 2: Transformers are limited by their expenditure of bounded compute per input instance, e.g. the finite context window and finite precision, thus cannot simulate a UTM, whose tapes are infinitely long (e.g. [79, 31, 80]).*

First, no finite physical system, such as a human brain or a personal computer, can solve problems of infinite size. This limitation naturally extends to the simulation of a UTM, which assumes infinitely long tapes. Therefore, when discussing whether a Transformer can simulate a UTM, the correct interpretation should follow the framework of the logical circuit model [81], specifically: "a uniform family of Transformers can simulate a UTM." In other words, for any arbitrarily large tape length $\ell$, there exists an efficiently constructible Transformer capable of simulating a UTM with tape length $\ell$.

In this context, while an exact simulation of a UTM is impossible, a sufficiently large Transformer can approximate its behavior to an arbitrarily high degree of accuracy. This scalability ensures that Transformers, much like circuits, can address increasingly complex problems within practical computational limits.

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

# A  Background on Turing Machines

**Turing machines.** Turing machines (TMs) are a mathematical model of computation. A $k$-tape TM is defined as a tuple $\langle \Sigma, \bot, Q, q_{start}, F, \delta \rangle$ where (i) $\Sigma$ is a finite tape alphabet including a blank symbol $\bot$, (ii) $Q$ is the finite set of states containing initial state $q_{start}$, (iii) $F \subseteq Q$ is a set of halting states, and (iv) $\delta$ is a transition function $(Q \setminus F) \times \Sigma^k \to Q \times (\Sigma \times \{L, S, R\})^k$.

Throughout this paper, we will assume $\Sigma = \{0, 1, \bot\}$ for simplicity and with loss of generality.

**Probabilistic Turing Machines**. A probabilistic Turing Machine (PTM) is a Turing machine with an additional read-only coin tape full of independent and uniformly random coins.

The PTM model is potentially more powerful than the deterministic Turing machine (DTM) model. An example of a computational problem that can be solved in polynomial time by a PTM but still not known how by a DTM is the polynomial identity testing problem (PIT) (see, e.g. [81]). In fact, it is a central question in complexity theory, well-known as the BPP $=$?P problem, whether any decision problem solvable by a polynomial-time PTM can also be solved by a polynomial-time DTM.

We say that a (deterministic or probabilistic) TM $T$ is *oblivious* if the tape head movements of $T$ running on input $x$ depend only on the input length $|x|$. That is, $T$ makes the same sequence of head movements for all inputs $x$ of the same length. [82, 83] proved that: for every multitape DTM $T$ running in $O(t(n))$ time, there is an equivalent oblivious two-tape DTM $T'$ that runs in $O(t(n) \log t(n))$ time. Furthermore, as observed by [84], this result also holds for all relative Turing machines, including PTMs. Specifically, for any multitape PTM $T$ running in $O(t(n))$ time, there is an equivalent oblivious two-tape PTM $T'$ running in $O(t(n) \log t(n))$ with an additional ready-only coin tape. So, w.l.o.g., in this paper unless otherwise specified, whenever we refer to DTMs or PTMs, we refer to two-tape oblivious DTMs or PTMs respectively.

**Universal Turing Machines**. A universal Turing machine (UTM) is a TM that can simulate the execution of every other (deterministic or probabilistic) TM $T$ given $T$'s description as input. Specifically, we encode PTMs as Boolean strings in a prefix-free way. A UTM is a PTM $U$ that takes the concatenations of the encoding of a PTM $T$ and an input $x$, and outputs the (possibly randomized) $T(x)$. UTMs capture the notion of a "general-purpose programmable computer", which is a single machine that can be adapted to any arbitrary task provided an appropriate program is loaded. We remark that the parameters of a UTM, such as alphabet size, number of states, and number of tapes are fixed, though the TM being simulated could have much more parameters.

# B  Proof of Theorem 2

*Proof.* We first adapt $T$ by introducing a lazy sampling of the coin tape. The coin tape is initially empty, filled with blank symbols $\bot$, and will be assigned random coins on the fly during execution. At one step, if $T$ reads a blank symbol $\bot$ from the coin tape, it first tosses a fair coin and writes the result on the coin tape at the current head position. Note that the adapted $T$, denoted by $T'$, runs for at most $2t(n)$ steps, since each original step may include an additional coin-tossing operation.

Next, we demonstrate how a transformer can simulate $T'$ with at most $2t(n)$ CoT steps. We adapt the proof of Theorem 2 in [20]. For the $i$-th step of $T'$, let $h_i^\tau \in \mathbb{Z}$ and $\gamma_i^\tau \in \Sigma$ denote the head position and the content on tape $\tau$, and let $q_i \in Q$ denote the state. Let $\Delta := Q \times \Sigma^2 \times \{L, S, R\}^3$, and $\delta_i \in \Delta$ denote the log at the $i$-th step, indicating the state entered, symbols written, and directions moved. The crucial observation is that the tape contents at the current head positions can be reconstructed from the input $x$ and the previous logs $\delta_0, \delta_1, \cdots, \delta_{i-1}$.

As shown in [20], a Transformer can first obtain all arguments $(q_{i-1}, \gamma_i^1, \gamma_i^2, \gamma_i^3)$ for the transition function. Suppose tape 3 is the coin tape. If $\gamma_i^3 \neq \bot$, which means that the $i$-th step of $T'$ will be deterministic rather than the coin-tossing operation, then the Transformer computes $\delta_i = \delta(q_{i-1}, \gamma_i^1, \gamma_i^2, \gamma_i^3)$ with a feedforward net outputting the one-hot encoding of $\delta_i$. If $\gamma_i^3 = \bot$, which means that the $i$-th step is a coin-tossing operation, then the Transformer outputs the equally weighted linear combination of the one-hot encodings of $(q_{i-1}, \gamma_i^1, \gamma_i^2, 0, S, S, S)$ and $(q_{i-1}, \gamma_i^1, \gamma_i^2, 1, S, S, S)$. The vector outputted by the feedforward net is then processed by the finial token classification head, which is a softmax function. One can check that the Transformer exactly simulates the $i$-th step of $T'$. $\qquad\square$

## C    Two remarks on Thesis 1

**Remark 2.** *There is ongoing debate as to whether quantum computers falsify ECT. In particular, it is a central problem in quantum computing, well-known as the* $\mathsf{BQP} =?\mathsf{BPP}$ *problem, whether all decision problems solvable by polynomial-time quantum computers can also be solved by polynomial-time PTMs. If ECT is falsified by quantum computers, then a quantum variant of Transformer that can simulate universal quantum computers [85, 41, 86, 87] might be necessary to achieve AGI.*

**Remark 3.** *It is widely accepted that a human brain can be modeled as a complex computational system (say, a huge neural network) following classical physical laws, and thus can be simulated by a PTM [88, 89]. However, this traditional view of the brain as a classical system was challenged by [90, 91]: they argued that the brain utilizes quantum mechanical effects (e.g., quantum coherence or entanglement) for reasoning and recognition, and human consciousness is even non-algorithmic, though still lack empirical validation.*

## D    Data requirement in Section 4.3

We discuss the types and amounts of data required in our framework to approximate AIXI using two Transformers.

- **Format Flexibility:** Data can be multi-modal (e.g. text, images) and multi-task (understanding, reasoning, etc.), aligning with current LLM practices. In fact, such diversity is essential to achieve increasingly universal models [30]. To further enrich the diversity, we suggest incorporating the synthetic UTM data. This types of data complements human-generated dataset by providing coverage over algorithmic patterns that may be underrepresented in the existing dataset. Moreover, the UTM data also provide a theoretical guarantees of convergence to SI.

- **Quantity:** Extensive data remain crucial for teaching induction. For deduction, however, the model can self-generate data (e.g., proposing solutions in math tasks and refining them via verification feedback).

