# OpenReview forum: "Transformers Have the Potential to Achieve AGI"
_NeurIPS.cc/2025/Position_Paper_Track — Submitted to NeurIPS 2025 Position Paper Track_

### Official Review · Reviewer_Qxd4 · 2025-07-22

**Significance:** 4
**Presentation:** 3
**Rating:** 4
**Confidence:** 2

**Summary:**

This position paper argues that the Transformer architecture—by its demonstrated capacity to simulate probabilistic turing machines, its accord with the extended Church–Turing thesis, and also the potential to approximete theoratical constructs—has both the theoretical expressiveness and practical promise to ultimately achieve artificial general intelligence. The authors marshal a three‑part argument. First, Transformers can simulatees any computable process via chain‑of‑thought prompting. Second, by the extended Church–Turing thesis, any physically realizable intelligence can in principle be emulated by a Transformer. Third, Transformers offer tractable approximations of universal search and induction frameworks, and are composed into an environment modeler and action planner to approximete AIXI in practice.

**Strengths:**

This is an ambitious call that directly links Transformers to core AGI constructs, giving a bold, unified vision.

The paper provides formal theorems and detailed backgrounds and theoretical contexts for readers to understand.

The call is impactful, given that there are always heavy discussions on whether LLMs or any Transformer-based model is the correct direction of AI development to achieve AGI.

**Weaknesses:**

The most important concern is the practical gap between the call and the real implementation scenario. The analysis of the paper is way too heavy on theory, and there is almost no empirical evidence showing how to realize this potential at scale. Also, this dense formalism may alienate readers seeking intuitive understanding or real‑world application guidance. I highly recommend that the author break down a little bit of the theory and link it to the current development of transformer-based models, and this may help strengthen your argument and clarity.

**Questions:**

Following the concern about the weakness of the gap between theory and implementation, how would you empirically validate or suggest to other researchers that the two‑Transformer AIXI approximation—what tasks or benchmarks would you propose?

**Alternative Position:**

Yes, and alternative positions are trivial straw-man arguments

**Author Identification:**

No.

**Context:**

3

**Discussion:**

3

**Ethics:**

["NO or VERY MINOR ethics concerns only"]

**Position:**

Yes, the paper argues for or against a position related to machine learning.

**Support:**

1

**Thoroughness:**

3

---

### Official Review · Reviewer_7Gn4 · 2025-08-12

**Significance:** 3
**Presentation:** 3
**Rating:** 7
**Confidence:** 4

**Summary:**

arguments:
1. It is shown that transformers with Chain of Though (CoT) can simulate Probabilistic Turing Machines (PTMs) with a number of CoT steps linear to the steps taken by the corresponding PTMs.
1. Assuming the extended Church-Turing thesis and based on the previous argument PTMs should be able to simulate any physical system with a polynomial computation overhead. Furthermore, citing the affinity of transformers to connectionism and symbolicism it is argued that transformers should be good approximations of the specific case of human reasoning.
1. In the more specific situation of problem solving, formalized by Hutter's AIXI agent, based on the ability of transformers with CoT to simulate universal Turing machines and efficiently solve sequence prediction, they are presented as a good approximant it. A specific framework with an environment modeler and action planner transformer is provided as a practical setup.

Finally, two alternative views are considered:
- Transformers lack embodied experiences and hierarchical planning capabilities.
- Transformers are limited by their bounded compute.

**Strengths:**

Strengths:

1. The paper has a clear statement and reasoning
1. Given the difficulty to define AGI it is nice that two different approaches are considered, one with the extended Church-Turing thesis and one with Hutter's AIXI agent.
1. The topic lies in the heart of the field and independently of the success of the authors' arguments it will probably trigger many discussions and draw more attention to it.

**Weaknesses:**

1. The paper considers a polynomial overhead to be acceptable in terms of efficiency in the extended CT thesis and in the attention mechanism which in general is quadratic on the number of input tokens. This could be both a practical and theoretical issue since assuming an agent needs to constantly process input which is arriving on pace linear to its size, then this poses a limitation on the input and CoT size. This could be a topic by itself as one should consider the different approaches to avoid quadratic space and the limitations they pose to the computational power of the agent as well as a study on the physical systems. Nevertheless, it is an alternative position that the quadratic time attention could be an inherent limitation of transformers that may not be present on other physical systems.

1. The first alternative view which is based on the interview of Yan LeCun does not seem to directly question the capabilities of transformers. Instead, he argues specifically against using only LLMs to achieve AGI. Also the combination of agents mentioned in the interview which could lead us closer to AGI are comparable to the framework described in section 4.3 with the exception of the idea of using the transformers as UTMs.

**Questions:**

No questions.

**Alternative Position:**

Yes, and alternative positions are well-considered and named but not addressed

**Author Identification:**

Yes, multiple of the authors.

**Context:**

3

**Discussion:**

4

**Ethics:**

["NO or VERY MINOR ethics concerns only"]

**Position:**

Yes, the paper argues for or against a position related to machine learning.

**Support:**

3

**Thoroughness:**

4

---

### Official Review · Reviewer_A8Q8 · 2025-08-12

**Significance:** 3
**Presentation:** 3
**Rating:** 7
**Confidence:** 3

**Summary:**

The paper posits that the Transformer architecture is a viable path to AGI. It supports this strong claim with three main theoretical arguments: (1) Transformers are Turing-complete and can simulate probabilistic universal computers, enabling them to perform any computation, including meta-algorithmic tasks. (2) The Extended Church-Turing Thesis implies that if AGI is physically possible, a Transformer can in principle replicate it. (3) Transformers can serve as tractable, practical approximations of theoretically optimal but uncomputable AGI agents like Solomonoff's Inductor, Levin's Searcher, and Hutter's AIXI. The paper culminates in a novel proposal for a two-Transformer framework to approximate the AIXI agent.

**Strengths:**

Builds on principles from theoretical computer science and algorithmic information theory, providing a formal and rigorous framework for AGI discussions rather than relying solely on empirical evidence.

integrates insights from diverse research areas, linking Transformer computational theory with practical methods for approximating Solomonoff Induction (SI) and Levin Search (LTS), leading to a distinctive AIXI approximation proposal. I appreciated the multidisciplinary stance that this paper posits its arguments on.

Addresses alternative, rather contentious, viewpoints, including concerns about embodiment and limited computational resources, showing awareness of opposing arguments despite providing only brief counterpoints.

**Weaknesses:**

The paper's primary weakness is its failure to adequately address the vast gap between the theoretical capabilities of idealized "tranformers" and the practical realities of trained LLMs. It heavily relies on Turing completeness proofs without discussing their critical, and likely invalidating, assumptions for standard models (e.g., hard attention, arbitrary precision)

The big claim about a "single Transformer" was challenged by recent work distinguishing between fixed models (computationally equivalent to finite-state machines) and evolving lineages of models (computationally equivalent to interactive TMs with advice). I think the paper would be amiss without incorporating this nuance. Doing so would make the argument more precise and defensible in my opinion.

Being a position paper, the authors should add a section on broader impact and ethical considerations, especially around this contentious topic. The dual-use nature of AGI research should be acknowledged. The authors should should discuss what safeguards might be necessary for the powerful, general-purpose systems they envision.

**Questions:**

Further to my review of the weaknesses of the paper, I think I would be interested in knowing what safeguards do the authors think might be necessary for the solution that they have proposed? Do you think a more concrete evidence should be provided that transformers can function as an environment modeler. Many luminaries in the field have attested that LLMs lack this capability.

**Alternative Position:**

Yes, and alternative positions are well-considered and addressed by the argument

**Author Identification:**

No.

**Context:**

2

**Discussion:**

4

**Ethics:**

["NO or VERY MINOR ethics concerns only"]

**Position:**

Yes, the paper argues for or against a position related to machine learning.

**Support:**

3

**Thoroughness:**

3

---

### Note · Authors · 2025-08-26

**1-11 Submit Again:**

Probably yes

**1-1 Submission Process:**

4

**1-2 Next Year:**

We would welcome adding a formal rebuttal phase (an author–reviewer discussion period) to next year’s position-paper track.

**1-4 Interest:**

["Mentorship programs for early-career researchers"]

**1-5 Thoughtful:**

7

**1-6 Supportive:**

6

**1-7 Technical Aspects Versus Position:**

6

**1-8 Gate Keeping:**

9

**1-9 Camera Ready Changes:**

Here’s what we will change in the camera-ready version, grounded in the reviews:

1. On the simulation of TM by transformers. Clearly state the idealized setting (arbitrary precision, unbounded context window) in Theorems 1&2. Add a discussion on inference efficiency and quadratic attention, summarizing recent results (e.g., PENCIL, constant bit-size transformers are Turing-complete) and practical mitigations (linear/sparse attention, RAG).

2. On implementation of the two-transformer framework. Provide the pipeline: Action Planner (AP) maximizes the Bellman objective using the Environment Modeler (EM) predictive distribution in place of $\mu$. Detail EM training protocol: synthetic UTM + curated real-world data, next-token NLL as the objective, and continual on-policy updates.
   -  Benchmarks: For EM, use ForecastBench, FutureBench, ARC-AGI, short-program sequences. For AP, use MATH, GSM8K, Codeforces-style tasks. For closed-loop, use MiniWorld, Procgen, or NetHack with sample efficiency, regret/optimality gap, and quality–compute curves under strict per-step budgets.

3. Safeguards: (fact vs. value):
   -  EM=fact judgment: prediction-only, calibrated uncertainty, HITL triggers, non-actuating.
   -  AP=value judgments: specifying the utility function is a value judgment and raises alignment needs; add verify-before-execute, sandboxing/least-privilege, and tripwires/interruptibility.

4. Alternative view clarification. Clarify that the LeCun interview critiques LLM-only paths rather than Transformer capability, aligning with our modular, agentic proposal.

5. Limitations & future work. Acknowledge remaining gaps between theory and current systems; emphasize targeted domain data, compute-budgeted inference, and stronger empirical validation as near-term priorities.

**3-1 Review Response1:**

A8Q8

**3-2 Reaction To Review1:**

Thoughtfulness: 8. Accurately summarizes our three-part argument and raises substantive concerns: idealized assumptions; safeguards; and empirical evidence of EM.
Supportiveness: 7.
Focus: 7.
Gatekeeping: 9.

Due to space constraint, we response to the concerns about safeguard and evidence of EM.

**About safeguard:** We propose to organize the discussion based on the distinction between "fact judgments" and "value judgments".
1. Environment Modeler (the fact judgment part): The EM performs purely epistemic tasks—it predicts how the environment responds to actions. This is a matter of **factual accuracy**, not normative choice. Accordingly, our safeguards focus on reliability and calibration, rather than value alignment. Concrete methods include: training data curation, uncertainty calibration, HITL, etc.
2. Action Planner (the value judgment part): Recall that AP acts to maximize a utility function based on the environment feedback. The definition of the utility function is essentially a value judgment, and belong to the normative domain. Thus it raises direct alignment concerns. Concrete methods include: value alignment, verify-before-execute, etc.

**About evidence of EM:** Benchmarks like FutureBench [1] demonstrate that LLMs can predict future events with reasonable accuracy in structured domains. Additionally, recent work such as Google's Genie 3 [2] models shows promise in generating interactive environments, though we acknowledge these are still limited in scope and complexity. Furthermore, LLMs' in-context learning capabilities [3] demonstrate their ability to adapt to environmental patterns during inference, updating their internal representations based on observed data—a key requirement for effective environment modeling.

[1] Back to The Future: Evaluating AI Agents on Predicting Future Events.
[2] Genie 3: A new frontier for world models.
[3] Language Models are Few-Shot Learners.

**3-3 Review Response2:**

7Gn4

**3-4 Reaction To Review2:**

Thoughtfulness: 8. Careful reading of our claims; incisive critique on quadratic attention/polynomial overhead and the LeCun interview interpretation.
Supportiveness: 8.
Focus: 7
Gatekeeping: 9.

The following are our detailed responses to the reviewer's concern.
Concern 1: About the quadratic overhead.
Response 1:  We will make this issue explicit in the revision and cite it as an important future direction. We list some recent progress in this direction:
- Yang et al. [1] proposed the PENCIL framework, which incorporates a reduction mechanism to recursively erase intermediate CoT step. They proved that any $s(n)$-space, $t(n)$-time TM can be simulated by PENCIL  with a $O(s(n))$-long window and a $O(t(n))$-long CoT. Consequently, the computation time per token is reduced from $O(t(n))$ to $O(s(n))$.
- Li et al. [2] proved that any $s(n)$-space, $t(n)$-time TM can be simulated by a constant bit-size transformer with a $O(s(n))$-long window and a $O(t(n)\cdot s(n))$-long CoT. Besides, by carefully examining the construction, the computation time per token is $O(1)$.

In addition, several empirical approaches have also been proposed to mitigate the quadratic blow-up in practice, such as linear/sparse attention and retrieval-augmented memories.

[1] Yang et al. PENCIL: Long Thoughts with Short Memory. ICML 2025.
[2] Li et al. Constant Bit-size Transformers are Turing Complete. https://arxiv.org/pdf/2506.12027.


Concern 2: About LeCun's interview interpretation
Response 2: Thanks for pointing this out. We apologize for the imprecise wording and will revise it accordingly.

**3-5 Review Response3:**

Qxd4

**3-6 Reaction To Review3:**

Thoughtfulness: 5. Emphasizes the theory–practice gap and requests concrete implementation and benchmarks.
Supportiveness: 4. Borderline but constructive.
Focus: Heavily on implementation rather than the position itself.
Gatekeeping: 8.

The following is our response to the reviewer's concern about the implementation:
1. Pipeline:  In each cycle $k$, Action Planner (AP) solves the Bellman equation (i.e., the maximization problem in  Line 340) using the Environment Modeler (EM) as a surrogate for the unknown $\mu$.

2. Training  EM:
- Goal. Approximate the Solomonoff prior in the limit (noting it is uncomputable). Formally, make $\Pr[\mbox{EM generates }x_n\mid x_1,\cdots,x_{n-1}]$ approximate $M_U(x_n)/M_U(x_{n-1})$. (see Definition 2 for the definition of $M_U(x_n)$).
- Pre-training Data. Mixture of real-world data and the synthetic UTM-generated sequence.
- Pre-training Objective. The standard objective: minimize the next-token NLL.

4. Examples of benchmarks for EM:  ForecastBench, FutherBench, ARC-AGI, etc.

5. Construction of AP:
- Goal. To efficiently solve the Bellman maximization and generally explicit decision-making or planning tasks.
- Current LLMs have already shown surprisingly deduction reasoning abilities, such as GPT-OSS. We could choose such a model as the base model.
- To further improve the deduction reasoning ability, we propose the following approach based on Levin search: First, by leveraging human experience or prior problem-solving data, one can train a model that takes a problem description as input and outputs a learned prior over candidate programs. Then we execute Levin's universal search with the default prior $2^{-\ell(p)}$ replaced with this learned prior. The intuition is that the learned prior is expected to assign higher weights to more promising algorithms.

6. Examples of benchmarks for AP: MATH, MGSM, Codeforces, etc.

7.  Examples of closed-loop evaluation benchmarks: Miniworld, NetHack, Procgen, etc.

---

### Meta-Review · Area_Chair_ar5M · 2025-08-31

**Rating:** 6
**Confidence:** 3

**Strengths:**

In this paper, the authors argue that the Transformer architecture is a credible route to AGI. The paper advances this in three steps: first, showing that transformers can simulate probabilistic Turing machines (this is a new result extending [1]); second, by the extended Church-Turing thesis, any realizable intelligent system is emulable by a probabilistic Turing machine; and finally, Transformers can therefore serve as approximations to optimal but uncomptuable agents (e.g., Solomonoff induction, Levin search, and AIXI). The authors' then propose a concrete two-Transformer design—an “environment modeler” and an “action planner”—as an AIXI approximation.

The reviewers all enjoyed the paper. In particular, highlighting:
- The paper has a clear and ambitious thesis that is tied to core AGI constructs.
- The paper grounds the thesis in ideas from theoretical computer science, rather than focusing on anecdotes about or evaluations of impressive capabilities.

**Weaknesses:**

The main weakness identified by the reviewers it the possibly large gap between the theory presented by the authors and practice. For example, the link between idealized arguments to trained, finite-precision, resource-bounded Transformers is underdeveloped. Proofs require things like hard attention, unbounded precision and other assumptions that may not hold for deployed models.

Additionally, central to the authors' position is what appears to be a novel result (based on the discussion of related research) that transformers can simulate probabilistic Turing machines. Since this is a new result, it is essential that it is treated as such and so I question the fit of this paper for the position paper track. As written in the call of position papers, "Submissions to the main NeurIPS conference track emphasize original research and novel results. In contrast, submissions to the position paper track will be judged primarily on whether they present a compelling position that warrants greater exposure within the machine learning community (regardless of whether a reviewer agrees with the position)."

**Questions:**

Please see my discussion of the paper's weaknesses.

**Ethics:**

There are no ethical concerns raised by the reviewers.

**Thoroughness:**

2

---

### Decision · Program_Chairs · 2025-09-26

Reject